# Single-Exciton Photoluminescence in a GaN Monolayer inside an AlN Nanocolumn

**DOI:** 10.3390/nano13142053

**Published:** 2023-07-12

**Authors:** Eugenii Evropeitsev, Dmitrii Nechaev, Valentin Jmerik, Yuriy Zadiranov, Marina Kulagina, Sergey Troshkov, Yulia Guseva, Daryia Berezina, Tatiana Shubina, Alexey Toropov

**Affiliations:** Ioffe Institute, 26 Politekhnicheskaya, 194021 St. Petersburg, Russia; john_fzf@mail.ru (E.E.); nechayev@mail.ioffe.ru (D.N.); jmerik@pls.ioffe.ru (V.J.); zadiranov@mail.ioffe.ru (Y.Z.); marina.kulagina@mail.ioffe.ru (M.K.); s.troshkov@mail.ioffe.ru (S.T.); guseva.julia@mail.ioffe.ru (Y.G.); dariya.burenina@mail.ioffe.ru (D.B.); shubina@beam.ioffe.ru (T.S.)

**Keywords:** excitons, 2D nanostructures, gallium nitride, photoluminescence

## Abstract

GaN/AlN heterostructures with thicknesses of one monolayer (ML) are currently considered to be the most promising material for creating UVC light-emitting devices. A unique functional property of these atomically thin quantum wells (QWs) is their ability to maintain stable excitons, resulting in a particularly high radiation yield at room temperature. However, the intrinsic properties of these excitons are substantially masked by the inhomogeneous broadening caused, in particular, by fluctuations in the QWs’ thicknesses. In this work, to reduce this effect, we fabricated cylindrical nanocolumns of 50 to 5000 nm in diameter using GaN/AlN single QW heterostructures grown via molecular beam epitaxy while using photolithography with a combination of wet and reactive ion etching. Photoluminescence measurements in an ultrasmall QW region enclosed in a nanocolumn revealed that narrow lines of individual excitons were localized on potential fluctuations attributed to 2-3-monolayer-high GaN clusters, which appear in QWs with an average thickness of 1 ML. The kinetics of luminescence with increasing temperature is determined via the change in the population of localized exciton states. At low temperatures, spin-forbidden dark excitons with lifetimes of ~40 ns predominate, while at temperatures elevated above 120 K, the overlying bright exciton states with much faster recombination dynamics determine the emission.

## 1. Introduction

Ultrathin quantum wells (QWs) GaN/(Al,Ga)N with thicknesses of 1–2 monolayers (MLs) are the structures of choice for the development of ultraviolet (UV) light-emitting devices with operating wavelengths (λ) in the highly demanded UVC and UVB ranges [1,2,3,4,5,6,7,8,9,10,11,12,13]. In particular, when using GaN/AlN multiple QWs (up to 400 periods), electron-beam-pumped UVC emitters with maximum peak output optical powers of 50 W for λ = 265 nm and 10 W for λ = 238 nm were demonstrated [12]. A distinctive feature of GaN/AlN QWs with thicknesses of 1–2 MLs is the extreme two-dimensional (2D) confinement of excitons on the scale of an atomic layer, which leads to the dominance of the exciton radiation mechanism up to room temperature [13]. This circumstance has important but contradictory consequences for the emission process. On one hand, highly confined excitons are beneficial for efficient light generation due to the increased rate of exciton transitions, which increases the radiation quantum yield [14,15]. On the other hand, excitonic radiation suffers from optical selection rules that reflect the conservation of both momentum and spin.

Due to the momentum conservation requirement, the QW excitons with large wave vectors outside of the light cone are optically dark. The effective rate of radiative decay of an exciton at elevated temperatures is defined as the average value over the thermal distribution of all excitonic states; therefore, the lifetime increases linearly with temperature [16]. The width of the “radiative window”, which includes exciton states with wave vectors within the light cone, is proportional to the square of the exciton transition energy, which is especially large in nanostructures of wide-gap semiconductors, such as GaN and AlN, which are, therefore, most suitable for creating efficient room-temperature (RT) excitonic emitters [13].

The selection rules that arise from the spin conservation requirement are especially pronounced in the thinnest GaN/AlN QWs, since the extreme confinement of electrons and holes along the growth axis leads to a sharp increase in the electron–hole short-range exchange interaction. As a result, the fine structure of the lower excitonic levels consists of two groups of states with four levels in each group: the four lowest states are dark, and the four highest states are bright. The exchange splitting between dark and bright excitons increases in combination with a decrease in the QW width and exceeds 40 meV for a single GaN/AlN monolayer emitting around 235 nm [13]. Such an exciton fine structure results in specific photoluminescence (PL) kinetics: long-living dark states are predominantly populated at low temperatures, while short-living bright states are activated with increasing temperature. Thus, despite the “dark” nature of the ground exciton states, ultrathin GaN/AlN QWs are effective exciton emitters at temperatures in the order of 300 K and above. 

So far, exciton PL in GaN/AlN QWs has only been studied in as-grown planar heterostructures, when radiation is collected from relatively large areas determined by the diameter of the excitation spot on the order of tens of microns or more [13]. The observed PL spectrum is inhomogeneously broadened (up to ~200 meV and even more), which indicates an effective in-plane localization of excitons due to fluctuations in the QW thickness. In addition to quantum confinement along the growth direction, the exciton energy, in this case, is additionally affected by the lateral size of the localization region. These measurements made it possible to obtain data on the exciton properties averaged over a large ensemble of localized excitons, but did not make it possible to determine the true characteristics of individual exciton states. Isolation and spectroscopic studies of single localized excitons can additionally shed light on the recombination mechanisms involved and be an important step toward the development of single-photon sources that emit in the UV solar-blind spectral range [17,18,19].

In this work, we studied individual nanocolumns fabricated using epitaxial heterostructures with a single 1-monolayer-thick GaN/AlN QW, which made it possible to observe narrow emission lines of single localized excitons. Measurements of the corresponding spectra and PL kinetics confirmed that, at low temperatures, dark localized excitons predominate in the emission, while an increase in temperature leads to the appearance of the PL lines of bright excitons at higher energies with much shorter radiative lifetimes. The areal densities of localization sites are estimated, and the characteristics of individual localized excitons are determined.

## 2. Materials and Methods

Three experimental samples (A, B, and C) with nanocolumns were fabricated by etching heterostructures with single AlN/GaN/AlN QWs grown via plasma-assisted molecular beam epitaxy (MBE) on a *c*-Al_2_O_3_ substrate [3,20,21]. Firstly, relatively thick 1.6-micrometer-AlN buffer layers were grown using multi-step metal-modulated epitaxy (MME) to reduce the threading dislocation density to less than 5 × 10^9^ cm^−2^ in the QW regions [20]. Further growth was carried out in two modes used to change the substrate temperature. In the first “high-temperature” (HT) mode, which was used for samples A and B, the buffer layer was grown at a substrate temperature Ts = 780 °C; the growth temperature was then lowered to 690 °C, and single QWs were immediately grown via the method described in our previous work [21]. Figure 1a shows the QW structure with this buffer. In the second “low-temperature” (LT) mode, which was used for sample C, after lowering the temperature, a 120-nanometer-thick AlN layer was grown via MME, and after that process occurred, a QW was formed, as shown in Figure 1b.

In both regimes, after growing the QWs and 5-nanometer-thick AlN covering layers, the temperature was raised to 780 °C, and the upper AlN 1-micrometer-thick layers were grown via the same method as the lower buffer layers. All layers in the structures were grown under metal (Me)-enriched conditions at flux ratios of Al/N = 1.1 and Ga/N = 2, which ensured continuous 2D growth in all samples. Moreover, Me-rich growth made it possible to determine the nominal QW thickness (*w*) using AlN growth rates calibrated under these conditions. The structures differed in this parameter, which was determined based on the opening time of the source of plasma-activated nitrogen during the growth of GaN QWs. In samples A, B, and C, the values of *w* were 1.1, 1.5, and 1.8 MLs, respectively. The correctness of this approach in determining the nominal thicknesses of QWs was verified in our previous works using high-angle annular dark-field scanning transmission electron microscopy (HAADF-STEM) [13,20,21]. In addition, these thicknesses were estimated via X-ray diffraction measurements of the multiple QW structures grown under similar conditions [20,21].

The nanocolumns were fabricated in two stages, which are schematically shown in Figure 1. In the first stage (Figure 1c), columns with diameters of 1–6 μm were fabricated using conventional photolithography and reactive ion etching (RIE) of a planar heterostructure. A combined capacitive and inductive discharge in a mixture of reactive gases BCl_3_/Ar was used in this process. In the second stage, the lateral size of the columns was reduced via wet etching in a 10% KOH solution at a temperature of 60–90 °C. This process is illustrated in Figure 1d. The two-step etching procedure allowed us to obtain very thin nanocolumns while eliminating a large number of defects typically associated with reactive ion etching. The diameters of the fabricated nanocolumns at the height of the GaN QWs were found in the range of 50–5000 nm. Selected images of individual nanocolumns, which were obtained using scanning electron microscopy (SEM) (CamScan S4-90FE, Cambridge, UK), are shown in Figure 2.

The fabrication of nanocolumns had two goals. Firstly, the small diameter of the nanocolumn led to the fact that only a small number of exciton localization centers were located inside it, which made it possible to observe narrow lines associated with the emission of individual localized excitons. In addition, the cylindrical nanocolumn played the role of an optical nanoantenna, facilitating the extraction of light from the semiconductor structure into the far optical field [22,23,24].

The radiative properties of both the as-grown heterostructures and individual nanocolumns were studied using PL spectroscopy implemented in a confocal optical scheme, with an intermediate magnified image used for visual control of the detection area (see Appendix A). The sample was fixed in a He-flow microcryostat, which made it possible to vary the sample temperature from 5 to 300 K. To excite PL, the fourth harmonic of a Ti-sapphire laser (Mira-900 with a harmonics generator, Coherent, Santa-Clara, CA, USA) was used, operating in a pulsed mode with a pulse repetition period of 13 ns and a pulse duration of 120 fs. The resulting pump wavelength was 215 nm, which corresponded to quasi-resonant excitation of the QW. Laser radiation was applied to the sample through an interference filter (RazorEdge 224 nm, Semrock, Rochester, NY, USA) and a reflex objective lens (LMM-40X-UVV, Thorlabs, Newton, NJ, USA). The average pump power density was approximately 1 W/cm^2^, and the excitation spot size was about 10 μm. When measuring the PL in the backscattering geometry, the interference filter prevented the laser radiation reflected from the sample from entering the spectrometer. 

The time-integrated PL spectra were recorded using a spectrometer (Acton SP2500, Princeton Instruments, Trenton, NJ, USA) with a diffraction grating of 1800 grooves/mm and a cooled CCD array (Pylon, Princeton Instruments, Trenton, NJ, USA). When operating in the first diffraction order optimal for the spectrometer used, the typical spectral resolution was ~0.2 nm. Using the second order of diffraction and minimizing the slits of the spectrometer, it was possible to improve the spectral resolution to 0.046 nm, albeit at the expense of sensitivity [25,26]. At a wavelength of 235 nm, this figure corresponded to ~1 meV. Time-resolved PL kinetics was measured using time-correlated single-photon counting, which is a sensitive technique used to record low-level light signals with picosecond resolution that is based on the detection of single photons of a periodic light signal, the measurement of the detection times, and the reconstruction of the waveform from the individual time measurements. The signal from a single-photon photomultiplier (PMC-100-4, Becker & Hickl GmbH, Berlin, Germany) was applied to a time-correlated single-photon counting module (SPC-130, Becker & Hickl GmbH, Berlin, Germany). A pin photodiode (PHD-400, Becker & Hickl GmbH, Berlin, Germany) was used for synchronization with the pulses of the fundamental harmonics of the Ti-sapphire laser. The time resolution of the setup, which was determined as the full width at half maximum (FWHM) of the response function, was ~140 ps.

The decay of low-temperature emission from single excitons confined in semiconductor QDs is typically biexponential [27]. The biexponential law is also characteristic of the PL decay kinetics observed in single GaN monolayers in AlN [13]. Therefore, when analyzing the measured decay curves, we approximated them by the sum of two decaying exponents:I(t) = A_1_·exp[−(t − t_0_)/τ_1_] + A_2_·exp[−(t − t_0_)/τ_2_],(1)
where t_0_ is the excitation time (the time when the excitation laser pulse reaches the sample), t is a delay time (the time interval between t_0_ and the moment of registration), τ_i_ is the decay time constant of *i*th PL component, and A_i_ is the corresponding amplitude. The experimental data were fitted using the least squares method based on the modeling function (1) with five fitting parameters: A_1_, A_2_, τ_1,_ τ_2_, and t_0_. To fit the PL decay curves over the maximum time interval (both before and after the excitation time), we took into account the contributions of 50 previous excitation pulses. To ensure the most accurate determination of the decay constant (τ_1_) of the fast PL component, convolution of the modeling function with the measured response function was used. An illustration of the fitting of the decay curve of a bright exciton is shown in Appendix A.

## 3. Results and Discussion

### 3.1. Time-Integrated Photoluminescence Spectra

Figure 3 shows the PL spectra measured in as-grown planar heterostructures A, B, and C. All structures have a broad PL band in the range of 230–280 nm, which is generally consistent with the nominal thicknesses of GaN/AlN QWs [2,11,28]. As previously reported for such samples, an oscillating structure is superimposed on the PL contours due to light interference in the entire heterostructure, including the thick AlN buffer layer [13]. Changing the excitation power in a wide range from 0.04 to ~20 W/cm^2^ (by a factor of 500) did not lead to noticeable changes in the shape of either the spectrum or PL decay curves (see Appendix A). These power densities are lower than those noted in typical experiments with single-photon sources based on single GaN quantum dots [26].

The maximum of the PL band in structure A with *w* = 1.1 ML is found at 238 nm (5.2 eV photon energy), which slightly exceeds the range of 230–235 nm, which is usually attributed to the radiation in a GaN/AlN QW with a nominal thickness of 1 MLs. The FWHM of this band, which is about 100 meV, is the smallest of all three structures. For structure B with *w* = 1.5 MLs, the PL band is about three times wider than that of structure A. It completely covers the wavelength range of 250 to 260 nm, and it is usually associated with the emission of GaN/AlN QWs with thicknesses of 2 MLs. The tails of this band propagate towards both shorter wavelengths (up to ~240 nm) and longer wavelengths (up to ~280 nm). These two structures were grown using the same HT-AlN buffer layers. 

Structure C differs from structures A and B as it uses a different buffer layer: the LT-AlN buffer. The PL band of this structure is the widest of all structures, being about 400 meV at half maximum. Its maximum is around 253 nm (photon energy 4.85 eV), though the tails extend to 260 and 230 nm toward longer and shorter wavelengths, respectively. Thus, the emission spectrum of this structure effectively overlaps the typical emission ranges of GaN/AlN QWs with thicknesses of both 1 MLs and 2 MLs. These data indicate a strong influence of the initial AlN topography on the morphology of GaN QWs, as briefly described in [12]. 

In line with the results of [13], the decrease in the integral PL intensity upon increasing temperature from 5 to 300 K for all samples is very modest. In particular, for sample C, which is most suitable for studying single excitons, the integral PL intensity drops from liquid helium to room temperature by only 10%.

Figure 4a shows the PL spectra obtained in individual columns of different diameters D, which were fabricated from structure A. For the thickest column with D~5 μm, the PL spectrum is represented by a broad band, which is practically equivalent to the PL spectrum in the as-grown planar heterostructure. Reducing the diameter leads to a gradual fragmentation of the smooth contour into many individual narrow lines. For sufficiently thin nanocolumns of less than ~300 nm in diameter, these narrow lines are well separated from each other, which makes it possible to estimate their total number in the spectrum. A similar behavior was observed in nanocolumns made from all three heterostructures. The surface density of such lines, which were obtained in a number of nanocolumns as the ratio of the number of lines to the cross-sectional area of the column, is shown in Figure 4b as depending on the diameter D. The surface density is practically independent of both the initial heterostructure and the column diameter, and it scatters between 500 and 1200 lines/μm^2^. If we assign each line to the radiation of one localized exciton, the average distance between the localization sites is between 30 and 45 nm.

For all structures, the nominal thickness of GaN/AlN QWs is below the critical thickness of GaN pseudomorphic growth on AlN, which was reported as being more than 2 MLs [29,30,31,32]. Therefore, such ultrathin QWs do not show a transition to three-dimensional Stranski–Krastanov growth, which is widely used to form self-organizing quantum dots (QDs) [18,19,29,30,31,32,33,34,35]. Instead of the strain-induced QDs, the observed localization mechanism can be attributed to fluctuations in the QW thickness and/or composition. Indeed, a non-uniform spatial distribution of the mean GaN content was previously observed within a GaN/AlN QW with a nominal thickness of 1.5 ML using HAADF-STEM [12,21]. The structure was characterized as being an array of regions visible on HAADF STEM images as regions of different brightness levels, where the local thickness varies from 1 to 2 MLs. The characteristic lateral size of fluctuations in the GaN content is estimated to be about 10 nm. The origin of this morphology was associated with the growth in GaN on AlN terraces with an equilibrium atomic step height of 2 ML [12].

It appears tempting to attribute the observed narrow PL lines to the emission of excitons confined within flat GaN regions that are 2-monolayers-thick and surrounded by AlN, which can be considered as “quantum disks”. In this model, the difference in radiation wavelengths can be explained based on the difference in the lateral quantum confinement of electrons and holes in disks with different lateral sizes. This explanation is in reasonable agreement with the lower-energy part of the spectra being around 250 nm, as recorded in structures B and C. The extreme confinement of excitons inside a 1–2-monolater-thick QW implies a strong decrease in the 2D exciton Bohr radius, compared to bulk material, to ~1.5 nm [36], which is equivalent to only a few MLs (1 ML is 0.259 nm for GaN). The characteristic lateral size of the GaN-enriched islands, which is more than 10 nm, found using HAADF STEM in [12] is almost an order of magnitude larger than the exciton Bohr radius, which implies the presence of negligible confinement energy. Therefore, the corresponding wavelength should be close to the radiation wavelength for an ideal QW with a thickness of 2 ML, which is actually about 250 nm.

This model, however, can hardly explain the very high confinement energy corresponding to states that emit in the same structures at shorter wavelengths. Assuming that the emission photon energy of a GaN/AlN QW with a nominal thickness of 2 ML is about 4.8 eV [13,28], the observation of narrow lines around 5.1–5.2 eV suggests that the confinement energy generated due to the lateral quantum confinement reaches 300–400 meV. In fact, the observed emission photon energies of about 5.2 eV are much closer to the energy of photons emitted by a GaN/AlN QW with a nominal thickness of 1 ML, which is about 5.3 eV (radiation wavelengths in the range of 230–235 nm) [13,28]. Therefore, the corresponding exciton localization sites are relatively shallow potential wells with different exciton localization energies. 

Taking into account the very small Bohr radii of extremely confined excitons, these sites can be considered to be clusters, including a very limited number of atoms, with a vertical size of 2–3 MLs, located inside a GaN/AlN quantum well with an average thickness of 1 ML. It is very likely that the states in all three samples responsible for low-temperature radiation with wavelengths shorter than ~240 nm are excitons localized at similar sites. Such small clusters are almost impossible to identify via any methods of transmission electron microscopy due to the inevitable averaging of the signal over the thickness of the prepared specimen. Useful information about the possible configurations of such clusters might be obtained from first-principal energy calculations, which are still lacking. On the other hand, as we will show in the next subsection, temperature-dependent and time-resolved PL studies can provide a comprehensive understanding of the nature of individual localized excitons of this type. 

Among all the samples, the nanocolumns in sample C are best suited to study such individual excitons due to the maximum spectral bandwidth of the emission. Provided that the densities of narrow PL lines that make up the entire band are approximately the same in all three samples (see Figure 4b), the widest spectrum refers to the lowest density of lines recorded in a fixed spectral range. Since we are concentrating on studies of excitons localized in 1-monolayer-thick QWs that emit around 235 nm, sample C is also preferred because this spectral range corresponds to the short-wavelength tail of the emission band, where the density of narrow PL lines must be further reduced.

### 3.2. Temperature-Dependent and Time-Resolved Photoluminescence

According to the temperature behavior, all observed narrow PL lines can be divided into two groups: the intensity of some lines (the first group) decreases with increasing temperature, while the intensity of other lines (the second group) shows an anomalous temperature dependence, i.e., an increase with increasing temperature. Lines of the second type are more often observed in the high-energy part of the entire spectrum; therefore, they can be attributed to the emission of bright localized excitons, the expected energy of which for a 1-monolayer-thick GaN/AlN QW is approximately 40 meV higher than the energy of the corresponding dark exciton [13]. Next, the lines of the first type can be naturally attributed to the emission of dark localized excitons. As we show below, measurements of the PL kinetics with time resolution convincingly confirm this assumption.

In some regions of the wide PL spectrum with a relatively low density of narrow lines, which were measured in fairly thin nanocolumns of structure C, we could directly observe pairs of lines whose intensity exhibited opposite temperature dependences. Figure 5a shows an example of such a pair, where two lines recorded near to 235 nm are separated by an energy gap of 37 meV, which fully corresponds to the values of the emission wavelength and exchange splitting previously reported for excitons in a single ML of GaN in AlN [13].

The low-energy line in such pairs, which are assigned to a spin-forbidden dark exciton, decays with increasing temperature and practically disappears above ~120 K. It is natural to explain this behavior based on the temperature filling of the overlying bright exciton states, which leads to a redistribution of the exciton population in favor of bright excitons. The overlying line in the pair is negligibly weak at low temperatures. This outcome is the natural behavior of bright excitons in GaN MLs due to the absence of the population of these states in thermal equilibrium. The observed signal is only associated with the recombination of the non-equilibrium exciton population immediately after the pulsed excitation. An increase in temperature gives rise to an equilibrium thermal population of bright states and, consequently, to a progressive increase in the PL intensity. At a temperature of about 60 K, the radiation intensities of the dark and bright excitons are equalized, while at higher temperatures, the radiation of the bright exciton dominates in the spectrum. Above temperatures of about 120 K, the PL lines of individual excitons in our samples overlap with each other. This circumstance limits the maximum operating temperature of promising single-photon emitters based on samples of this type. Achievement of the higher operation temperatures relies on the fabrication of samples with lower density of the sites of exciton localization.

Careful consideration, in Figure 5a, of the low-energy line attributed to a dark exciton reveals, at low temperatures, some internal fine structure that cannot be fully resolved due to the limited spectral resolution of the grating spectrometer optimized for first-order diffraction. Using second-order diffraction and minimizing slits of the spectrometer, the fine structure of this line measured at 10 K can be confidently resolved, as shown in Figure 5b. 

Previously, the fine exciton structure of an ideal narrow GaN/AlN QW with the wurtzite crystal structure was theoretically studied using group-theory analysis [13]. The symmetry of such a QW is described based on the point symmetry group *C_3v_*. It was shown that the eight ground exciton states are associated with the valence-band states that originated from the p_x_ and p_y_ orbitals of the constituent bulk material, which were, in turn, oriented in the plane of the QW grown on the c-oriented substrate. The states that originated from p_z_ orbitals were pushed via strong size quantization to higher energies by several hundred meV. These eight ground states were separated by an electron–hole short-range exchange interaction between two groups of four states in each group, as shown in Figure 6. In the states of the upper group, the involved electron and hole spins are antiparallel, and the corresponding transitions are dipole-allowed (bright), preserving the total spin in the system. In the four lowest exciton states, the electron and hole spins are parallel, and the states are dipole-forbidden (dark).

In addition, each group of states splits into two doublets due to the spin–orbit interaction. The bottom doublets are formed via the A valence sub-band of bulk GaN, and the top states are formed via mixed B and C sub-bands. Thus, the splitting between doublets should be of the order of the A–B exciton splitting in GaN, which is ~7.8 meV. 

Both pairs of bright states transform under the $\Gamma_{3}$ representation of the group *C_3v_* [13,37,38]. These states are optically active in $\sigma^{+}$ and $\sigma^{-}$ polarizations and have the in-plane dipole moments necessary for radiative recombination, which is responsible for the PL signal recorded from the surface of the structure. The lower doublet of dark states also has $\Gamma_{3}$ symmetry, while the other two states belong to the $\Gamma_{1}$ and $\Gamma_{2}$ representations. The latter states can be split due to spin–orbit mixing using remote bands. We note that the state with symmetry $\Gamma_{1}$ can be active only in the z polarization. This observation means that the contribution of such states to the PL signal from the structure surface must be vanishingly small. Transitions involving states $\Gamma_{2}$ are forbidden by symmetry.

All four lower exciton states are spin-forbidden. However, they may still be optically active due to mixing with remote bands. For example, dark states with $\Gamma_{3}$ symmetry can mix weakly with bright states, and states with $\Gamma_{1,2}$ symmetry can mix with states originating from p_z_ orbitals due to the hole spin–orbit interaction [13]. Phonon- and impurity-assisted processes, as well as intersite hopping, can also contribute to optical activity involving dark states. In any case, we expect that the average radiative lifetime for spin-forbidden dark excitons should be much longer than for bright exciton states. 

The fine structure observed in the PL spectrum of a dark localized exciton (Figure 5b) agrees, in some aspects, with the theoretical description of the fine structure of an exciton carried out for a 1-monolayer-thick GaN QW. Indeed, three narrow lines (their width is defined by the spectral resolution of the spectrometer that operates in the second-order diffraction mode) are perfectly resolved. In addition, a small shoulder is observed on the low-energy side of the multiplet, which can be attributed to the emission of the fourth exciton state. This finding exactly corresponds to the number of states theoretically predicted for the ground multiplet of spin-forbidden excitons in a narrow GaN QW with *C_3v_* symmetry. Moreover, the total spectral width of the multiplet (~10 meV) reasonably matches the expected exciton splitting defined by the spin–orbit interaction. This agreement supports the assumption that the main contribution to the size quantization of confined electron and hole states is made through the z-confinement in the 1–2 monolayer-thick GaN, while the influence of the confinement in the lateral directions is much weaker. However, lateral confinement reduces the states’ symmetry, canceling the strict selection rules expected for an ideal QW with a wurtzite crystal structure.

The low-temperature PL kinetics in all initial samples studied (A, B, and C) perfectly reproduces all features reported previously in [13]. At each energy inside the inhomogeneously broadened band, the PL signal is a superposition of a great number of lines that originate from the emission of both dark and bright localized excitons. Correspondingly, the PL decay is biexponential, consisting of a slowly decaying part that arises from dark excitons and a rapidly decaying contribution, which is a non-equilibrium emission of bright excitons. By studying the kinetics of individual localized excitons, we can expect to separate these two contributions.

Figure 7a shows the PL decay curves being spectrally integrated over all lines of dark excitons or the line of a bright exciton, as indicated in Figure 5a. Due to the relatively weak optical pumping, the detected signal level was rather weak and amounted to about 10 and 100 photons per second for the bright and dark exciton lines, respectively. The decay curve obtained for the line of the bright exciton (5.296 eV) differed only slightly from the response function of the setup, which did not allow us to accurately determine the decay-time constant. However, assuming that the decay is strictly exponential, the convolution of the modeling function with the measured setup response function made it possible to roughly estimate the value of τ_1_. Figure 7b shows fitting results obtained at fixed decay times τ_1_ of 5, 60, 100, and 140 ps. As can be seen from a comparison of the simulation and experimental curve, the decay constant τ_1_ definitely does not exceed 100 ps. The insert in Figure 7b shows the dependence of the sum of squared residuals (SSR) on the value of τ_1_, which reflects the accuracy of the fitting. The best fit was obtained with τ_1_ = 60 ps; however, for a more reliable determination of this time constant, a better temporal resolution is required. The estimated value of τ_1_ reflects both the kinetics of radiative recombination of a bright exciton and the characteristic rate of reaching equilibrium, which at 10 K is mainly determined via exciton relaxation to dark states, which is accompanied by the emission of acoustic phonons [27].

The decay time of the PL line attributed to the dark exciton (E = 5.259 eV) is much longer. As we can see in Figure 7a, the PL signal does not completely decay between two excitation pulses, which are separated by an interval of 13 ns. Assuming strictly monoexponential decay and taking into account the contributions of a number of successive excitation periods, we simulated the decay curve and estimated the corresponding decay time constant to be 39 ns, which is three times greater than the interval between exciting laser pulses. Such a long radiative lifetime favors the easy saturation of the states and, consequently, the observation of all emission lines in the multiplet. On the other hand, we have never observed such an internal structure for PL lines associated with the emission of bright excitons. These lines are always single, which can be explained based on the rapid relaxation of the excitation inside the multiplet at low temperatures and the overlap of the broadened lines at elevated temperatures.

Thus, the observed difference in the PL decay rate by a factor of ~500 convincingly confirms the assignment of dark and bright excitons, which was first made on the basis of the PL temperature dependences and the resolved fine structure of dark exciton states. Such accurate measurements of both the temperature dependence and the PL kinetics in individual narrow lines were only possible for a few lines, since the density of localization sites is typically high, while most neighboring lines overlap spectrally, especially at elevated temperatures. However, the general behaviors of all studied samples in the spectral range of 230–240 nm are similar: lines, or bundles of overlapping lines, whose intensity decreases with temperature exhibit very slow decay at low temperatures with a time constant ~40 ns, while in regions where the PL spectrum is characterized by opposite temperature dependence, there is always a large contribution from the signal decaying on a time scale that is faster than the time resolution of our setup.

## 4. Conclusions

In this paper, we study the kinetics of exciton recombination in MBE-grown 1-monolayer-thick GaN/AlN QWs at the level of single localized excitons. By measuring the emission of an ultrasmall region of a QW enclosed in a cylindrical nanocolumn, we found that the PL spectrum consists of a set of narrow emission lines attributed to individual excitons localized via fluctuations in the width of the QW. The overall width and shape of the inhomogeneously broadened spectrum measured in a planar as-grown heterostructure depend on the details of epitaxial growth, though the nature of the involved exciton states for the radiation wavelength range of 230–240 nm is basically the same in all of the structures studied. The PL spectrum at low temperatures is dominated by spin-forbidden dark localized excitons with a long decay time of about 40 ns. As the temperature increases, this radiation is quenched in favor of the emission of spin-allowed bright exciton states, whose contribution gradually increases following the temperature-induced increase in their population. This study elucidates the dominant mechanisms of radiative recombination in UV-C light-emitting devices based on GaN/AlN heterostructures and paves the way for the development of single-photon devices that operate in the solar-blind spectral range based on the emission of single localized excitons.

## Figures and Tables

**Figure 1 nanomaterials-13-02053-f001:**
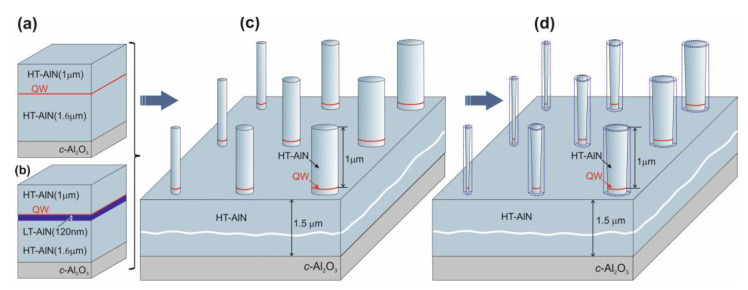
Planar heterostructures with a GaN/AlN QW grown (**a**) on “high-temperature” (HT) AlN and (**b**) “low-temperature” (LT) AlN buffer layers. Schematic representations of the fabrication of nanocolumn arrays after (**c**) the first stage (initial reactive ion etching) and (**d**) wet etching in 10%KOH.

**Figure 2 nanomaterials-13-02053-f002:**
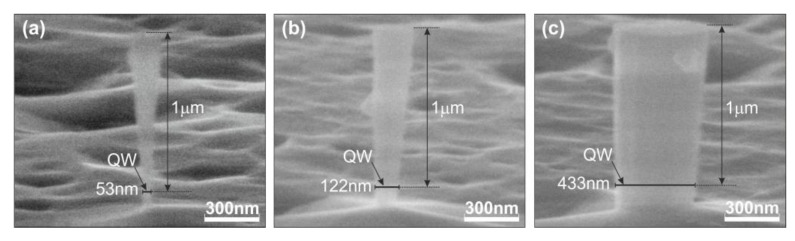
SEM images of nanocolumns with diameters at the height of the GaN QW of 53 nm (**a**), 122 nm (**b**), and 433 nm (**c**).

**Figure 3 nanomaterials-13-02053-f003:**
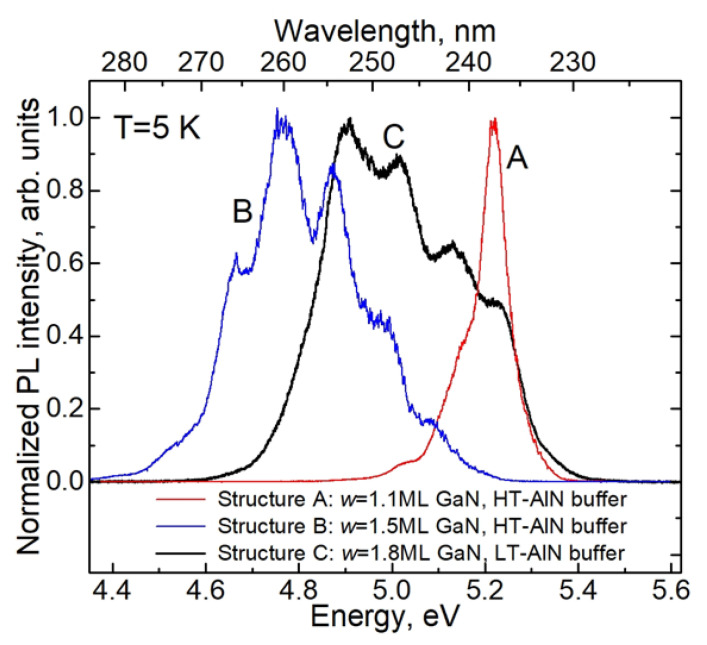
Normalized time-integrated PL spectra measured at 5 K in as-grown heterostructures A, B, and C.

**Figure 4 nanomaterials-13-02053-f004:**
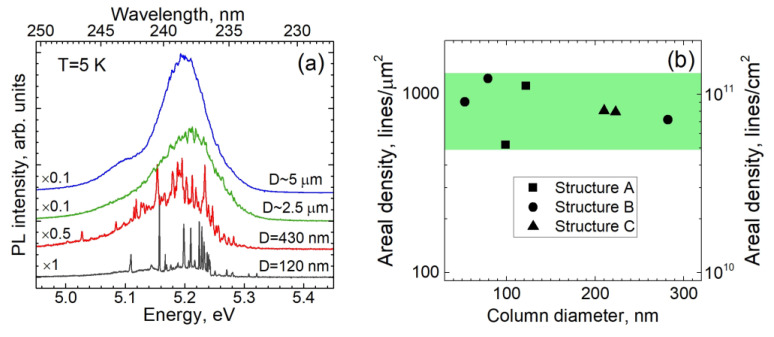
(**a**) Time-integrated PL spectra measured at 5 K in the columns of sample A with different diameters D, which were measured at the height of the GaN QW, as indicated in the figure. (**b**) Density of narrow lines, which were obtained for columns of different diameters, fabricated from different structures, as a function of D. The green bar indicates the range of surface density of narrow lines observed in all structures studied.

**Figure 5 nanomaterials-13-02053-f005:**
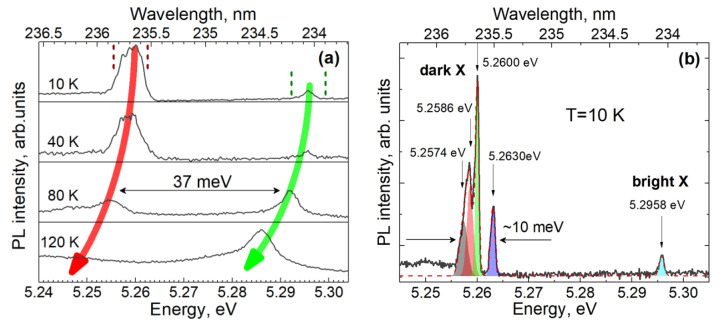
(**a**) A fragment of the time-integrated PL spectrum measured at different temperatures in a 210-nanometer-thick nanocolumn, which is made of structure C, that demonstrates a pair of lines associated with the emission of a dark exciton (lower energy line) and a bright exciton (higher energy line), which are laterally localized in the same place. The vertical dashed lines show the wavelength ranges in which time-resolved PL was measured at 10 K. Red and green arrows are drawn to guide the eye. (**b**) PL spectrum of dark and bright localized excitons measured at 10 K in the second order of diffraction with improved spectral resolution. The shaded areas and dashed lines show the results of spectrum approximation based on the sum of 4 Gaussian contours and 1 Gaussian contour for the dark exciton and bright exciton lines, respectively.

**Figure 6 nanomaterials-13-02053-f006:**
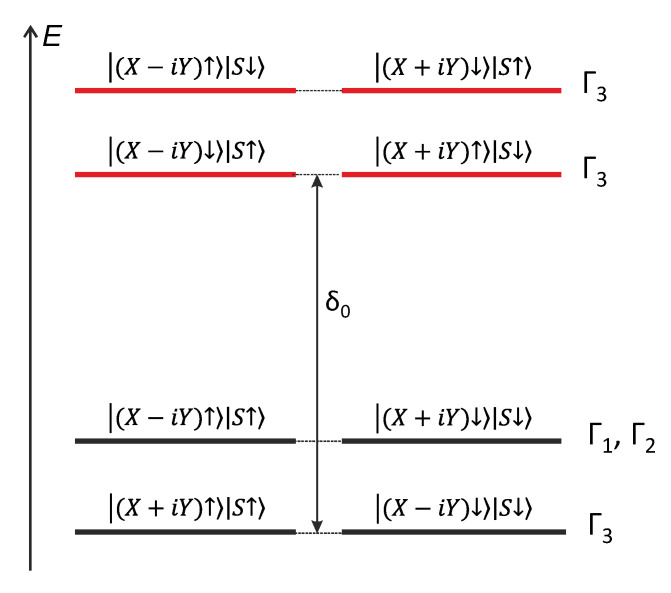
Fine structure of ground exciton states in an atomically thin GaN QW and representations of states in the point symmetry group *C_3v_*. The first ket vectors describe the hole orbital functions and spin, while the second ket vectors describe the electron spin. X, Y, and S denote p- and s-type orbitals, respectively. δ_0_ is the energy gap associated with the short-range electron–hole spin-exchange interaction.

**Figure 7 nanomaterials-13-02053-f007:**
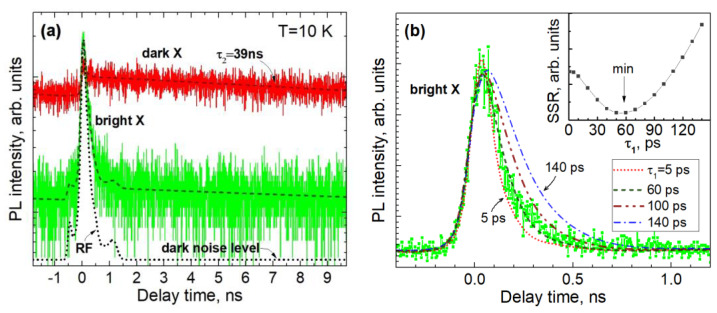
(**a**) Time-resolved PL decay curves of dark and bright localized excitons measured at 10 K (red and green curves, respectively). Dashed lines represent the results of simulation. The dotted line shows the setup response function with a dark noise level. (**b**) The measured PL decay curve of a bright exciton compared to model functions, with a decay time constant τ_1_ equal to 5, 60, 100 and 140 ps. The insert shows dependence of the sum of squared residuals of the fitting versus the fast time constant τ_1_. The best fitting parameters of the model functions are A_1_ = 0.075 and τ_1_ = 60 ps for the bright exciton line and A_2_ = 0.0027 and τ_2_ = 38.87 ns for the dark exciton line.

## Data Availability

The data presented in this study are available on request from the corresponding author.

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
