# Peer review of "Single-Exciton Photoluminescence in a GaN Monolayer inside an AlN Nanocolumn"

_nanomaterials, 2023, doi:10.3390/nano13142053_

Round 1

Reviewer 1 Report

The title may sound intriguing, but it might lead to an overestimation of its content or significance. I suggest the inclusion of the term "nanocolumn" in the title and adjusted the term "single-exciton."

Regarding the Materials and Methods section, it would be beneficial for readers to have a clearer understanding of the photoluminescence setup. It is recommended to provide a comprehensive description of the optical setup, including details about the spectrometer, the fast PMT (photomultiplier tube), and the TCPSC (time-correlated single-photon counting) board used for time-resolved photoluminescence (PL) measurements. Additionally, it would be helpful to include a reference explaining the use of second-order diffraction to enhance spectral resolution. In terms of data analysis for time-resolved PL, it is important to explain the deconvolution process. Line 172 mentions a temporal resolution of ~160 ps, and it would be valuable to clarify whether this refers to the bandwidth of the response functions or the resolution achieved after deconvolution. Please provide a proper discussion on this matter.

In Figure 3, where the authors present the spectral changes of samples A, B, and C, it would be beneficial to discuss not only the intensity changes but also the results of fluorescence lifetime measurements.

For Figure 7, it is recommended to add baselines to the three datasets. Specifically, the red line (dark X) appears not to reach a fully relaxed equilibrium within the ~13 ns interval. Considering the temporal resolution of 160 ps mentioned earlier, the decay constant of ~70 ps in line 389 suggests that the fitting value may not be meaningful. Furthermore, the slow decay of ~40 ns is approximately three times longer than the pulse interval of ~13 ns. Please report all fitting results, include residual plots with chi-square values, and specify which function was used for the fit.

The authors claimed the presence of four different dark excitons and one bright exciton in their PL spectrum at 10 K. It would be valuable to know if time-resolved PL measurements were performed for all five excitons. In Figure 7, only the bright and dark excitons are displayed. Please provide an explanation of the wavelengths at which time-resolved PL was detected for the dark and bright excitons. Additionally, include the results for all excitons in your measurements.

Author Response

We are grateful to the Referee for a careful reading of the manuscript and useful remarks. Below we provide answers to each remark of the Referee and describe the respective improvements made in the manuscript.

The title may sound intriguing, but it might lead to an overestimation of its content or significance. I suggest the inclusion of the term "nanocolumn" in the title and adjusted the term "single-exciton."

ANSWER:

We follow the Reviewer’s recommendation. The title now reads as

“Single-exciton photoluminescence in a GaN monolayer inside an AlN nanocolumn”

In this variant, firstly, the system under study (a GaN monolayer in a nanocolumn) is correctly described. Secondly, the too general term " spectroscopy" is excluded. However, we keep the term “single-exciton” because it is usually used to emphasize the role of a single exciton in some effect (see, e.g. [1]).

 [1] V.I. Klimov et al., Single-exciton optical gain in semiconductor nanocrystals, Nature 447, 441–446 (2007)

Regarding the Materials and Methods section, it would be beneficial for readers to have a clearer understanding of the photoluminescence setup. It is recommended to provide a comprehensive description of the optical setup, including details about the spectrometer, the fast PMT (photomultiplier tube), and the TCPSC (time-correlated single-photon counting) board used for time-resolved photoluminescence (PL) measurements.

ANSWER:

In the respective paragraph of the Materials and Methods section (lines 170 - 181), we provide specifications for all devices listed in the Referee remark as well as several additional experimental details.

Additionally, it would be helpful to include a reference explaining the use of second-order diffraction to enhance spectral resolution.

ANSWER:

We introduce two new references: Ref. 25 and Ref. 26 (line 175). Ref. 25 is a handbook, providing theory of the operation of a grating monochromator. Ref. 26 is a paper providing an example of using second-order diffraction to enhance spectral resolution.

In terms of data analysis for time-resolved PL, it is important to explain the deconvolution process. Line 172 mentions a temporal resolution of ~160 ps, and it would be valuable to clarify whether this refers to the bandwidth of the response functions or the resolution achieved after deconvolution. Please provide a proper discussion on this matter.

ANSWER:

For definiteness, we refer in the revised version to the bandwidth (full width at half maximum) of the response function, which is 140 ps. This is mentioned in the respective paragraph of the Materials and Methods section (line 182). More details of the deconvolution process are provided in the same paragraph (lines 182-190).

In Figure 3, where the authors present the spectral changes of samples A, B, and C, it would be beneficial to discuss not only the intensity changes but also the results of fluorescence lifetime measurements.

ANSWER:

The PL lifetime measurements in the samples identical to samples A, B, and C were measured and analyzed in all details in Ref. 13. In the current paper, we focus on the studies of photoluminescence of single localized excitons. Therefore, in the revised version, we note that PL kinetics in all initial heterostructures studied (A, B, and C) perfectly reproduces all features reported previously in [13] and emphasize the difference between this behavior and the kinetics of individual localized excitons (lines 411-418).

For Figure 7, it is recommended to add baselines to the three datasets.

ANSWER:

Figure 7a is modified so that the setup response function is shown with a dark noise level. This baseline is the same for all datasets.

Specifically, the red line (dark X) appears not to reach a fully relaxed equilibrium within the ~13 ns interval. Considering the temporal resolution of 160 ps mentioned earlier, the decay constant of ~70 ps in line 389 suggests that the fitting value may not be meaningful. Furthermore, the slow decay of ~40 ns is approximately three times longer than the pulse interval of ~13 ns. Please report all fitting results, include residual plots with chi-square values, and specify which function was used for the fit.

ANSWER:

We didn’t quite understand the Referee’s remark that “the red line (dark X) appears not to reach a fully relaxed equilibrium within the ~13 ns interval.” As we mention in lines 432-434 of the revised manuscript: “One can see in Figure 7 (a) that the PL signal does not completely decay between two excitation pulses, which are separated by an interval of 13 ns.” However, fitting performed in the model described in lines 182-190 readily shows a single-exponential character of the decay that is also mentioned (lines 432-437). In our opinion, there are no specific signatures of not reaching equilibrium for the system demonstrating pure exponential decay. Nevertheless, this is quite possible, since we measured the decay of emission integrated over all dark excitons. Certain saturation, which is a kind of nonequilibrium, can take place within the muliplet of dark excitons. This is mentioned in the revised manuscript (lines 438-439).

We agree with Referee that an accurate determination of the decay time constant for the emission of a bright exciton is hardly possible for the conditions of our experiment. However, we can reliably estimate it as being less than 100 ps by comparing the simulated and experimental curves. This comparison is shown at the revised Fig. 7b. The simulation is described in the paragraph in lines 419-431 and in the Fig. 7 caption.

The authors claimed the presence of four different dark excitons and one bright exciton in their PL spectrum at 10 K. It would be valuable to know if time-resolved PL measurements were performed for all five excitons. In Figure 7, only the bright and dark excitons are displayed. Please provide an explanation of the wavelengths at which time-resolved PL was detected for the dark and bright excitons. Additionally, include the results for all excitons in your measurements.

ANSWER:

It was not possible to perform time-resolved PL measurements for the separate lines of dark excitons due to their obvious spectral overlapping. In the revised Fig. 5 we directly show the spectral ranges where the time-resolved PL was measured. This is described in lines 419-420.

Reviewer 2 Report

The article “Single-exciton photoluminescence spectroscopy in a GaN 2 monolayer in AlN” presents some interesting research on monolayer GaN layers. But major revision is needed for suitability to the journal.

Major comments:

1) The thicknesses of wells must be experimentally verified by TEM analysis. Time-correlated equipment must be detailed. What is the error of bright exciton lifetime?

2) Bright and dark exciton binding energies and decay times must be calculated theoretically with equations provided.  Provide exciton decay times and binding energies in a table for clarity.

3) What is the quantum efficiency of the bright excitons? Authors state about single exciton devices operation at room temperature but the spectra at RT are not shown in Fig. 5a, are bright excitons activated?

Minor comments:

Fig 4b – point is outside figure

Author Response

We are grateful to the Referee for a careful reading of the manuscript and useful remarks. Below we provide answers to each remark of the Referee and describe the respective improvements made in the manuscript.

Comments and Suggestions for Authors

The article “Single-exciton photoluminescence spectroscopy in a GaN 2 monolayer in AlN” presents some interesting research on monolayer GaN layers. But major revision is needed for suitability to the journal.

Major comments:

  • The thicknesses of wells must be experimentally verified by TEM analysis. Time-correlated equipment must be detailed. What is the error of bright exciton lifetime?

ANSWER:

TEM analysis of the monolayer-thick layers is a difficult task, which is out of the scope of this paper. Moreover, it was done in our previous work (Refs 20 and 21 in the revised version) with similar samples. Therefore, in this paper we identify the QW nominal thickness relying on the technological calibration, made with the account of previous TEM studies and X-ray diffraction measurements of the multiple QW structures grown under similar conditions.  This is formulated in lines 112-116 of the revised manuscript.

The error of bright exciton lifetime is discussed in the revised version in lines 419 – 431 and in the modified Fig. 7b. We emphasize that an accurate determination of the decay time constant for the emission of a bright exciton is hardly possible for the conditions of our experiment. However, we can reliably estimate it as being less than 100 ps simply comparing the simulated and experimental curves. Time-correlated equipment is detailed in the last paragraph of the Materials and Methods Section.

  • Bright and dark exciton binding energies and decay times must be calculated theoretically with equations provided.  Provide exciton decay times and binding energies in a table for clarity.

ANSWER:

Theoretical calculation of the exciton binding energies and other exciton properties in a monolayer-thin QW is an extremely difficult task that is definitely out of the scope of this paper. In addition, the accuracy of these calculations is very low. Simple approaches like the kp theory are not valid here and one should apply first-principles calculations and solve the Bethe-Salpeter equation. We know only one paper, where authors tried to do that. In the revised version we refer this work to as ref. 35.    

  • What is the quantum efficiency of the bright excitons? Authors state about single exciton devices operation at room temperature but the spectra at RT are not shown in Fig. 5a, are bright excitons activated?

ANSWER:

Above about 120 K, the PL lines of individual excitons in our samples overlap with each other. This circumstance sets the limit to the highest operation temperature of prospective single-photon emitters based on the samples of this type. Achievement of the higher operation temperatures relies on the fabrication of samples with the lower density of the sites of exciton localization (lines 326 – 330 in the revised manuscript).

Note that the whole set of our experimental data does not allow determination of the quantum efficiency of the bright excitons.

Minor comments:

Fig 4b – point is outside figure

ANSWER:

Fig. 4b is corrected.

Reviewer 3 Report

The authors fabricated cylindrical nanocolumns 50 to 5000 nm in diameter from GaN/AlN single QW heterostructures grown by molecular beam epitaxy, using photolithography with a combination of wet and reactive ion etching. The kinetics of exciton recombination in MBE-grown GaN/AlN QWs has been studied at the level of single localized excitons. By measuring the emission of an ultrasmall region of a QW enclosed in a cylindrical nanocolumn, it is found that the PL spectrum consists of a set of narrow emission lines attributed to individual excitons localized by fluctuations in the width of the QW. The overall width and shape of the inhomogeneously broadened spectrum measured in a planar as-grown heterostructure depend on the details of epitaxial growth. This study elucidates the dominant mechanisms of radiative recombination in UVC light-emitting devices based on GaN/AlN heterostructures and paves the way for the development of single-photon devices operating in the solar-blind spectral range using the emission of single localized excitons.

I think that the work is well written, easy to follow. This is an interesting study with high quality experimental data and convincing conclusion. I’d like to recommend this paper for publication in the Nanomaterials.

Author Response

The authors fabricated cylindrical nanocolumns 50 to 5000 nm in diameter from GaN/AlN single QW heterostructures grown by molecular beam epitaxy, using photolithography with a combination of wet and reactive ion etching. The kinetics of exciton recombination in MBE-grown GaN/AlN QWs has been studied at the level of single localized excitons. By measuring the emission of an ultrasmall region of a QW enclosed in a cylindrical nanocolumn, it is found that the PL spectrum consists of a set of narrow emission lines attributed to individual excitons localized by fluctuations in the width of the QW. The overall width and shape of the inhomogeneously broadened spectrum measured in a planar as-grown heterostructure depend on the details of epitaxial growth. This study elucidates the dominant mechanisms of radiative recombination in UVC light-emitting devices based on GaN/AlN heterostructures and paves the way for the development of single-photon devices operating in the solar-blind spectral range using the emission of single localized excitons.

I think that the work is well written, easy to follow. This is an interesting study with high quality experimental data and convincing conclusion. I’d like to recommend this paper for publication in the Nanomaterials.

ANSWER:

We are grateful to the Referee for a careful reading of the manuscript and a high appreciation of our results.

Round 2

Reviewer 1 Report

The authors expressed their support for using the term "single-exciton" and claimed that their experiments were conducted within the single-exciton regime. They utilized a relatively strong excitation light power of approximately 1W/cm2 with a 10-micron spot size, as stated in the manuscript (lines 167-168). To provide further clarification, they requested excitation power-dependent time-resolved photoluminescence (PL) data to determine the specific conditions under which single-exciton generation occurred.

In the revised version of the manuscript, the authors requested more detailed information about the time-correlated single-photon counting (TCSPC) technique employed. They also asked for a reference to be included for Equation (1) (line 186) and for thorough explanations of the associated parameters. They expressed uncertainty about whether the terms "delay time" and "excitation time" accurately represented these parameters.

Furthermore, the revised manuscript mentioned that the full width at half maximum (FWHM) of the instrumental response function (IRF) in the time-resolved PL measurements was approximately 140 ps, which was close to the temporal resolution. The authors requested that the temporal resolution be explicitly stated in the measurements. They also inquired whether any additional analysis, such as deconvolution to account for the convolution between the PL data and IRF, was performed to enhance the temporal resolution. Additionally, they highlighted a discrepancy in line 440, where the reported fitted decay was obtained within 50-80 ps. Considering the temporal resolution of approximately 140 ps, it appeared unlikely that faster decay times could be resolved.

Moreover, the lifetime constant of approximately 39 ns exceeded the time interval of approximately 13 ns that was under consideration. The authors suggested fitting alternative models, such as an incomplete-decay model that accounts for fluorescence signals remaining from previous laser pulses, instead of relying on a simple multi-exponential decay. This approach would provide a better means of resolving longer lifetime components present in the data.

In the responses of the authors, they answered “It was not possible to perform time-resolved PL measurements for the separate lines of dark excitons due to their obvious spectral overlapping. In the revised Fig. 5 we directly show the spectral ranges where the time-resolved PL was measured. This is described in lines 419-420.” However,

Is it feasible to use time-resolved photoluminescence (TRPL) measurements to resolve the exciton fine structure, considering that in Figure 5 (b), the time-integrated photoluminescence (PL) spectrum at 10 K utilizes second-order diffraction to enhance the spectral resolution, resulting in an improved resolution of 0.046 nm? I don’t understand why they answered, “It was not possible to perform time-resolved PL measurements for the separate lines of dark excitons due to their obvious spectral overlapping.” Based on the experimental details, it seems to have resolved them.

Reviewer 2 Report

The revised paper is slightly corrected, but the device application limitations are not strictly described in the conclusions as the PL is weak at room temperature.

Round 3

Reviewer 1 Report

The authors of this manuscript present findings on the time-integrated and time-resolved photoluminescence in GaN/AlN nanocolumn well structures. Based on the research conducted, including fabrication and optical properties analysis, I recommend publishing the article in Nanomaterials after making some minor revisions outlined below:

1.    The reviewer's comments suggest publishing the excitation-power dependent time-resolved photoluminescence (TRPL) data in the supporting materials, even if it is not suitable for inclusion in the main manuscript.

2.    In terms of TRPL data analysis, the authors utilized a biexponential decay model, as far as my understanding goes. However, on line 195, the authors explain the term "deconvolution," which may mislead readers into thinking it refers to the deconvolution of the convoluted data between the instrument response function and the real fluorescence decay. This should be clarified to prevent any misunderstanding.

Overall, with these minor revisions, I believe the manuscript is suitable for publication in Nanomaterials.
